# Review of Research on Tire–Pavement Contact Behavior

Zhenlong Gong [1,2], Yinghao Miao [1,*] and Claudio Lantieri [2,*]

1 National Center for Materials Service Safety, University of Science and Technology Beijing, Beijing 100083, China; gongzl@xs.ustb.edu.cn
2 Department of Civil, Chemical, Environmental and Material Engineering (DICAM), University of Bologna, 40136 Bologna, Italy
* Correspondence: miaoyinghao@ustb.edu.cn (Y.M.); claudio.lantieri2@unibo.it (C.L.)

**Abstract:** This article presents the latest progress in research on tire–pavement contact behavior. Firstly, the tire–pavement contact characteristics and their influencing factors are summarized. Then, the measurement methods and theoretical research on tire–pavement contact behavior are reviewed, and the advantages and shortcomings of different methods are compared and analyzed. Finally, analysis in the field of pavement engineering is summarized based on contact behavior. This article suggests a few key research directions: Tire–pavement contact behavior is influenced by multiple factors; therefore, multi-physical field-coupling analyses need to be carried out. Tire–pavement contact tests are mostly static and non-standardized, and it is a future trend to develop high-precision, low-cost, and standardized instruments that can measure dynamic contact. Theoretical research models rarely involve environmental factors; a contact model of the tire, pavement, and environment needs to be constructed that can truly describe the contact process. There is a relationship between contact characteristics and pavement performance; pavement performance evaluation indexes need to be established based on tire–pavement contact characteristics in the future.

**Keywords:** tire–pavement; contact behavior; pavement engineering; measurement method; theoretical research; pavement performance





## 1. Introduction

The tire–pavement contact problem is one of the important problems in the field of pavement engineering [1–3]. Tire–pavement contact behavior is essential to understanding issues such as pavement skid resistance [4,5], noise [6], rolling resistance [7], and driving safety and comfort [8,9]. Tire–pavement contact involves tribology [10], acoustics [11], contact mechanics [12,13], tire dynamics, and other multidisciplinary fields [14] and has attracted widespread interest in the field of pavement engineering. The tire–pavement contact process is also the process of pavement skid resistance deterioration [15,16], pavement disease formation [17], and tire performance decline [18]. Therefore, the study of tire–pavement contact behavior is of great significance for improving pavement performance, enhancing pavement life, and enhancing vehicle energy saving and environmental protection [19,20]. Furthermore, only with a clear understanding of the interaction between the tire and the pavement can we obtain real contact stress and pavement load distribution data. Only then can we design more economical and durable pavements as well as carry out friction analyses.

Currently, there are two main types of research methods for tire–pavement contact behavior: One is contact experimental research, which analyzes tire–pavement contact characteristics through tests, but these tests are difficult to perform, and only in recent years has there been related research. Some researchers use pressure sensors [21], carbon paper [22], and pressure-sensitive materials [23] to test the pressure distribution. The second type of research method is theoretical research, in which researchers analyze tire–pavement contact behavior through mathematical models. Later, with the development

of computer technology, many researchers have studied the interaction between tires and pavements through numerical analysis techniques based on theoretical research [24–26]. The finite element analysis method can build tire and pavement models quickly and efficiently and analyze the tire–pavement contact accurately and cost-effectively, but there are differences between simulation and reality [27]. Experimental research can truly reflect the tire–pavement contact behavior, but these tests have problems such as high costs and external environmental influences [1,13]. Although tire–pavement contact behavior and its influence on pavements have been understood to some extent, contact characteristics are still controversial, and the influence law on the pavement is still unclear. Some researchers have carried out numerical simulations and experimental research on the tire–pavement contact behavior, but there is a lack of detailed and complete summary and generalization.

The objective of this article is to conduct a comprehensive review of the latest progress in the study of tire–pavement contact behavior. In order to achieve this objective, tire–pavement contact characteristics, factors influencing contact behavior, contact measurement methods, and theoretical research are reviewed. Firstly, the research progress on tire–pavement contact behavior is examined based on both geometric characteristics and mechanical characteristics. Secondly, the influencing factors of contact behavior are analyzed based on three aspects—the tire, pavement, and contact environment. Then, experimental research on tire–pavement contact behavior and measurement methods are summarized, and the advantages and disadvantages of different measurement methods are evaluated. Finally, tire–pavement models and computer simulations are reviewed, and the advantages and disadvantages of different research approaches are discussed. This article also outlines the analysis of tire–pavement contact behavior to study noise, skid resistance, rolling resistance, and pavement damage. Future research directions are also envisioned. The framework of this article is shown in Figure 1.

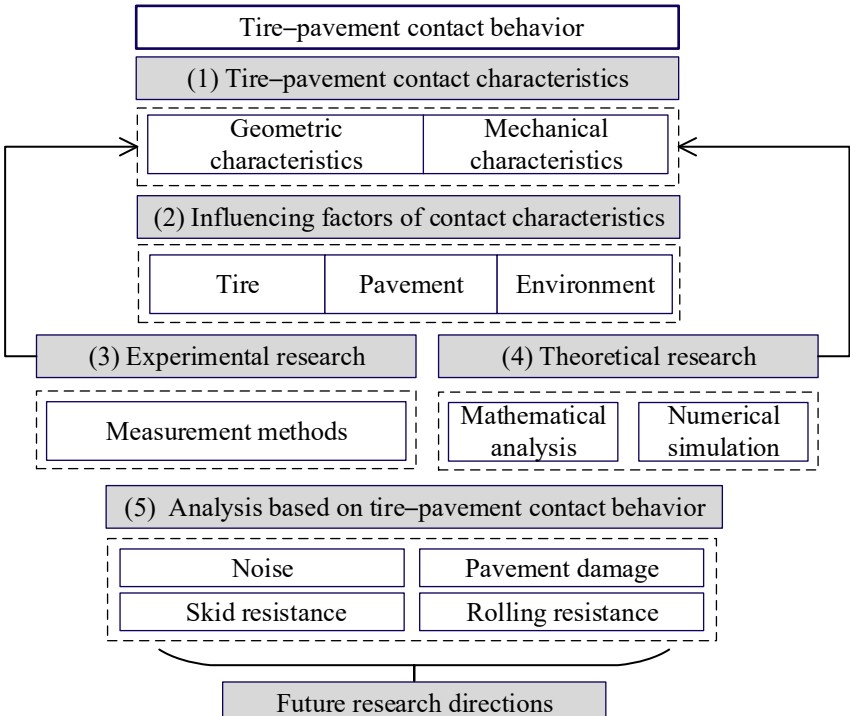

**Figure 1.** Article framework.

## 2. Tire–Pavement Contact Characteristics

Tire–pavement contact behavior is understood to help optimize tire design and pavement structure design. This can help improve pavement performance and the overall performance of tires. Researchers have characterized tire–pavement contact behavior

through tire–pavement contact characteristics [2,28]. Tire–pavement contact characteristics are mainly geometric characteristics defined according to the contact area and mechanical characteristics defined according to the stress distribution [29,30]. The definition, understanding, and application of tire–pavement contact characteristics vary among researchers because of differences in the study subjects and methods.

*2.1. Geometric Characteristics*

The geometric characteristics of tire–pavement contact mainly refer to the contact area and contact shape. There is no clear definition of contact characteristics in the field of pavement engineering; however, the geometric characteristics of tire–pavement contact are summarized and defined in the field of tires, and "Tire Terms and Their Definitions" (GB/T 6326-2005) [31] define the tread contact length (*L*), tread contact width (*W*), coefficient of contact (*L/W*), contact area (*C-A*), and footprint area (*F-A*), as shown in Figure 2.

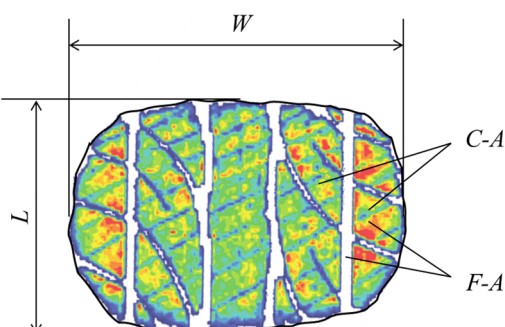

**Figure 2.** Geometric characteristics of tire–pavement contact.

The tire–pavement contact area is usually assumed to be circular or rectangular with uniform pressure distribution. This can simplify mechanical calculations and reflects tire–pavement contact behavior to a certain extent, and is widely used in pavement design [32]. However, measurements and modeling data show that the vertical stress distribution at the tire–pavement interface is not uniform, and the contact area is not regularly circular [29] or rectangular [33,34]. Tire pattern and pavement texture influences are ignored in this assumption, so researchers have conducted extensive research on contact geometric characteristics [35,36]. Pillai et al. [37] used the tire footprint to determine the tire–pavement contact area and obtained two simple predictive equations for tire–pavement contact area based on tire deflection and tire and wheel dimensional parameters found on the tire sidewall, as shown in Equations (1) and (2), with an error between the calculated and measured footprint areas within 15%. Ge and Wang [38] established a tire–pavement contact finite element model using ABAQUS to simulate the tire–pavement contact process and obtain a contact area equation based on tire pressure and wheel load, as shown in Equation (3). A quantitative expression of tire contact characteristics under vehicle load was obtained, as shown in Figure 3.

$$A = 2.17dr \tag{1}$$

$$A = 1.85d^{0.67}r^{0.33}(2r - s)/2a \tag{2}$$

where *A* is tire footprint area; *r*, *s*, and *a* are tire radius, wheel diameter, and aspect ratio, respectively; and *d* is tire deflection.

$$\delta = 0.1233p^{-0.9580}F^{0.9511} \tag{3}$$

where $\delta$ is tire footprint area, $10^3$ mm; *p* is tire pressure, Mpa; and *F* is wheel load, kN.

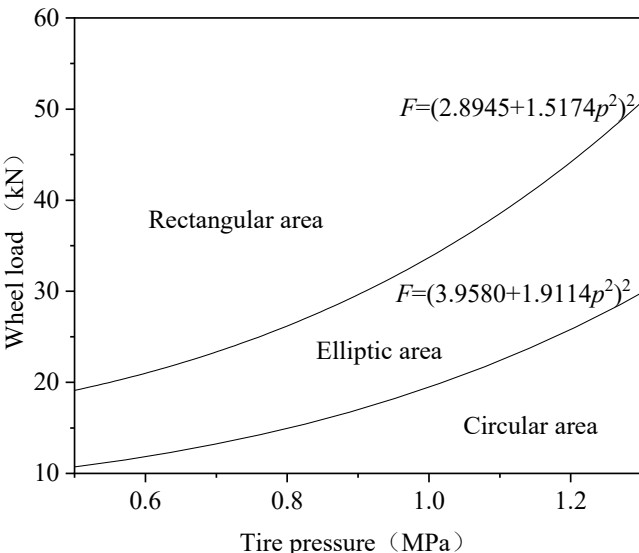

**Figure 3.** Distribution map of the tire contact area.

Tielking et al. [39] found that the contact area between the tire and the pavement increases as the wheel load increases, and the effect of a single wheel load on the contact area is lower. Based on experiments, Weissman et al. [40] found that the width of the tire footprint remained essentially constant due to the stiffness constraint of the tire side-wall, and that the wheel load mainly affected the length of the tire footprint. Yu et al. [2] conducted a static tire–pavement contact test using super-low-pressure (LLW) Fujifilm Holdings Corporation (FUJI) pressure film paper. As the mean profile depth (MPD) increases, the actual tire–pavement contact area decreases, and as the tire load increases, the contact area gradually increases. However, when the tire load exceeds 250 N, the tire–pavement contact area does not broadly change and even shows a slight decrease. This is mainly because the contact width becomes larger when the load is too large but the contact length is significantly reduced. In summary, it can be seen that the geometric characteristics of tire–pavement contact are related to the wheel load, pavement texture, tire type, etc. Research has mostly used the footprint method for static tire–pavement contact geometric characteristic analysis, lacking dynamic contact studies, while more direct simplification is used in pavement design.

### 2.2. Mechanical Characteristics

Tire–pavement contact creates stresses and forces. These tire–pavement contact stresses and forces provide the driving force when the vehicle accelerates, the lateral force when it rotates, and the braking force when it decelerates, and it can be seen that the contact behavior is directly related to vehicle performance [41] and pavement performance [42,43]. Therefore, it is significant to conduct contact mechanical characterization studies. When a tire is in contact with pavement, the tire is subjected to a complex load through the complex deformation of the tread and carcass. This complex load is generally described in the study of tire–pavement contact dynamics as the six tire forces, which are vertical force, longitudinal force, lateral force, aligning torque, lateral tilt moment, and rolling resistance moment. Some of these forces are shown in Figure 4. Due to the large deformation of a tire in mutual contact with pavement coupled with the environment and load, these forces are coupled with each other and affect each other [39,44].

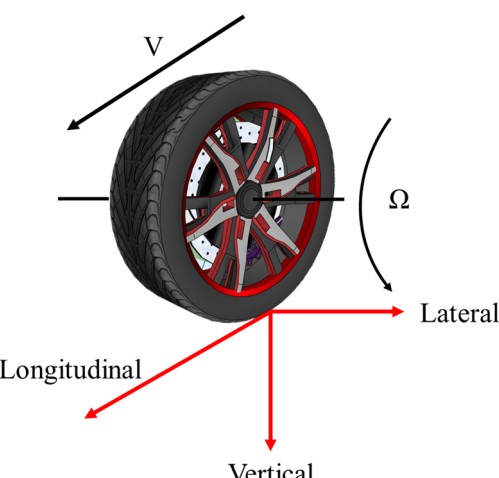

**Figure 4.** Schematic diagram of contact stress.

The vertical forces are of most concern in pavement studies. Marshek et al. [45] attempted to test the pressure distribution footprint of diagonal truck tires using pressure-sensitive films, and their results showed that the contact pressure was non-uniformly distributed and that the contact pressure at the tire shoulder was significantly greater than the tire air pressure. Wang et al. [46] analyzed the non-uniformity of vertical contact stresses and found that the non-uniformity of stress distribution decreases with increasing load but increases with increasing inflation pressure. Gong et al. [47] used a high-precision pressure sensor to test the tire–pavement contact pattern and stress under different forces, as shown in Figure 5. It can be seen that with an increase in force, the contact area and the region of stress concentration gradually increase. The distribution of vertical contact stresses is not uniform.

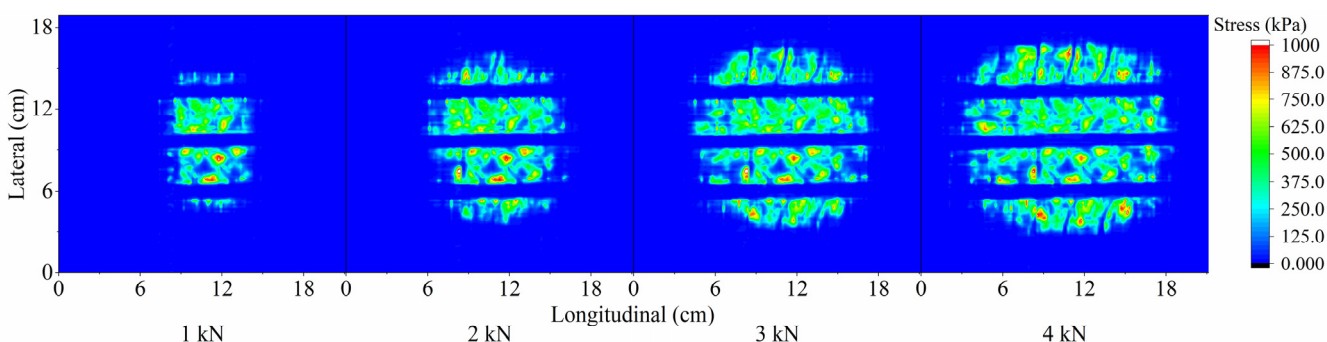

**Figure 5.** Contact stress distribution images.

For longitudinal forces, Tielking et al. [39] analyzed longitudinal contact stresses and vertical contact stresses as a function of vehicle speed. They found that vehicle speed had almost no effect on the vertical contact stress but had a significant influence on the longitudinal contact stress; the longitudinal stress on the pavement and the tire changed direction twice from the stationary state to the rolling state. Douglas et al. [48] tested the distribution of vertical and longitudinal pressures between the tire and the pavement under different tire pressures and loads. Their test was a full-scale laboratory test and confirmed Tielking's analysis of the contact stresses. For lateral forces, Hugo et al. [49] found that lateral forces can cause pavement cracking. Research on the aligning torque, lateral tilt moment, and rolling resistance moment has mostly been conducted in the field of vehicle engineering. In summary, it can be seen that the mechanical characteristics of tire–pavement contact are closely related to pavement performance and vehicle performance, and there are certain rules. Therefore, we can understand tire–pavement contact behavior better by characterizing these mechanical characteristics.

## 3. Factors Influencing Contact Characteristics

Many factors influence tire–pavement contact characteristics, but the main ones are tire factors, pavement factors, and environmental factors, as shown in Figure 6. Tire factors such as tire tread, type, and pressure all have an effect on contact area and contact stress. The pavement and the environment are equally important in influencing the contact characteristics.

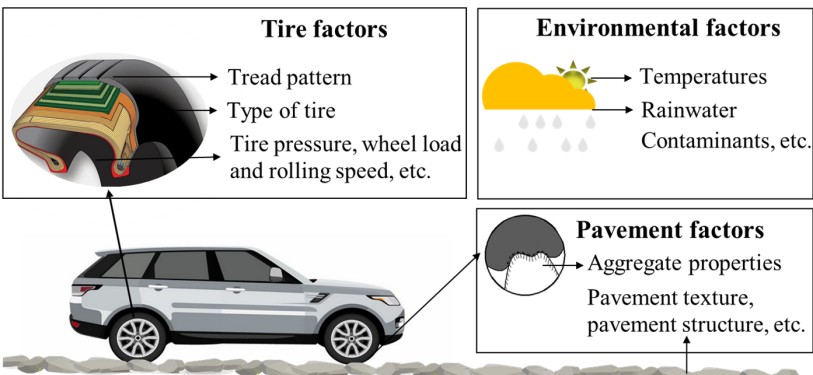

**Figure 6.** Factors influencing contact characteristics.

### 3.1. Tire

In the case of tires, the type of tire, tire pressure, tread pattern, wheel load, and the rolling speed of the tire all impact tire–pavement contact. Xie and Yang [30] analyzed the effect of tire type on contact stress by establishing a tire–pavement three-dimensional finite element model. The results showed that with the increase of vertical load, the position of the maximum contact stress between the diagonal tire and the pavement changed from the inner side of the tread to the outer side, while the maximum contact stress between the radial tire and the pavement always appeared in the middle of the tread. Under the same working conditions, the maximum contact stress between radial tires and pavement is greater than the maximum contact stress between diagonal tires and pavement. Furthermore, the change in lateral contact stress under different ribs of diagonal tires is smaller than that of radial tires. In this paper, the contact area of different tires was quantified, as shown in Figure 7. It can be seen that there are differences in the contact areas of different tire types, although this broadly conforms to the law that the larger the tire load, the larger the contact area. All the points in Figure 7 are linearly fitted, which achieves a good fitting effect. According to the 95% prediction band, the contact area of different tires also varies within a certain range.

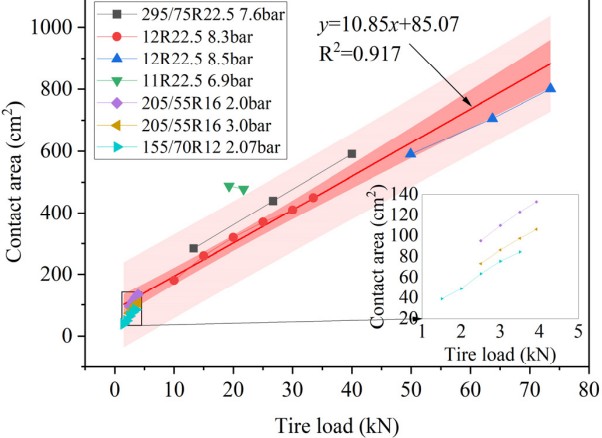

**Figure 7.** The contact area of different tires [3,21,29,50–52].

Muniandy et al. [36] selected seven different tread-pattern tires for footprint image analysis, and their results showed that the tread pattern affected the tire–pavement contact area second only to tire load. Machemehl et al. [53] used measured tire–pavement contact stress data to study the effect of tire pressure on pavements. Tire pressure was found to be significantly related to the tensile strain at the bottom of the asphalt concrete layer and the stress near the pavement surface for both thick and thin pavement structures. Wang et al. [54] analyzed the tire–pavement interaction by measuring contact stress for different loads and constant tire pressures. It was found that an increase in the wheel load mainly increased the contact stress at the edge of the tire and the corresponding shear strain and octahedral shear stress. Zheng et al. [3] established a three-dimensional tire–asphalt pavement contact model based on the Lagrangian algorithm and investigated the effects of internal inflation pressure and wheel load on the tire–asphalt pavement contact characteristics under static load and antilock braking system (ABS) states, respectively. The results show that a tire is more prone to wear at the shoulder position and tread-center area during emergency braking, and the center area has the highest contact pressure. With the same wheel load and internal inflation pressure, the mean contact pressure increases by 8.4%, and the contact area decreases by 7.7% when the tire is under ABS compared to static load. In summary, it can be seen that tire changes cause changes in tire–pavement contact behavior, and there are certain patterns of effect.

## 3.2. Pavement

Pavement factors include pavement structure, pavement texture, aggregate and asphalt characteristics, etc. Driving pavements are mainly asphalt and cement concrete pavements, which have significant differences. The Permanent International Association of Road Congresses (PIARC) classifies pavement textures into four categories—microtexture, macrotexture, mega texture, and unevenness—and tire–pavement contact studies are mainly conducted at the macrotexture and microtexture levels [55,56]. Pavement texture directly affects tire–pavement contact behavior. In addition, asphalt pavements are mainly composed of aggregates and asphalt, which inevitably affects tire–pavement contact behavior.

### 3.2.1. Pavement Type

Research on tire–pavement contact behavior has been more specific to one type of pavement, and few studies have compared the effects of asphalt and concrete pavements on contact behavior. Sheng et al. [57] used 3D printing technology to produce pavements with three different textures to analyze the effect of variations in pavement texture on stress distribution through numerical simulation. It was found that different textures lead to different locations of maximum stress distribution. He et al. [58] also used ABAQUS to simulate asphalt pavement–tire contact behavior, which differed from the results of Wei et al. This was mainly due to the difference in stiffness between asphalt and concrete pavements, which leads to different simulation results, especially for a pavement's internal stresses, which can show large differences. It can be seen that pavement type affects the stress distribution, and pavement topography directly affects the distribution of stress concentrations. Thus, most research on tire–pavement contact behavior focuses on pavement texture rather than pavement type.

### 3.2.2. Texture

Asphalt and cement pavements are shown in Figure 8; it can be seen that there is little difference in the appearance of the two pavements but notable differences in their textures [59]. Asphalt pavement texture is random and irregular, but the mean texture depth (MTD) of asphalt pavement for a specific gradation is stable. Cement pavements are very flat due to the fact that the grooves are kept in a specific pattern. Thus, the mean texture depth is broadly the same at different locations on a pavement. Texture differences can lead to different tire–pavement contact behavior. Yu et al. [2] used FUJI pressure film paper to test the differences in the tire–pavement contact area for different

MPDs, and the results showed that the pavement contact area decreased with increasing MPD. Gong et al. [47] analyzed the contact stress between asphalt pavement with different MTDs and tires and found that pavement texture had an effect on the contact behavior and showed a relationship with the MTD value. However, some researchers have ignored pavement texture during their studies, arguing that texture has a limited effect on contact stresses [32,60]. Zheng et al. [3] considered pavement texture during their modeling and analysis process and found that pavement texture had a significant effect on contact stresses.

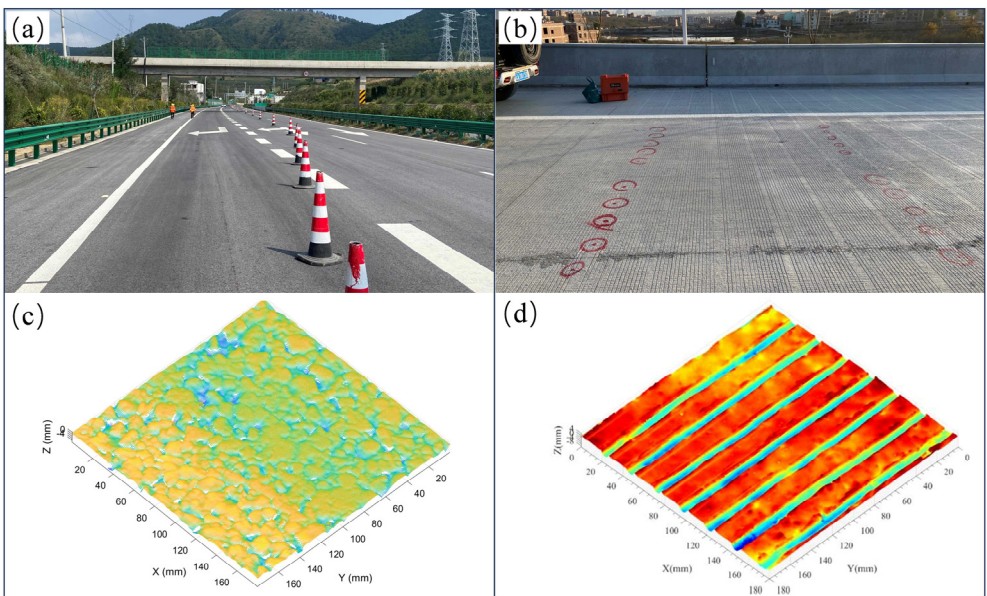

**Figure 8.** Asphalt and cement pavements: (**a**) Asphalt pavement; (**b**) Cement pavement; (**c**) Asphalt pavement texture; (**d**) Cement pavement texture.

Tire–pavement friction is determined by microtexture and macrotexture [61]. Most existing research is on macrotexture as it is more difficult to measure microtexture and, consequently, it is more difficult to analyze the effect of microtexture on tire–pavement contact behavior. At this stage, the impact on pavement performance is mostly determined by analyzing the morphological characteristics of the aggregates [62]. Sengoz et al. [63] determined the effect of different shapes of aggregates on the tire–pavement interface by experimentation. It was found that there is a relationship between the shape characteristics of the aggregates and the surface properties of asphalt concrete. However, their research did not examine tire–pavement contact behavior. Nevertheless, tire–pavement contact behavior has an impact on many pavement performance indicators, such as noise and skid resistance. Some relevant research has considered the effect of microtexture [64,65].

### 3.3. Environment

Tire–pavement contact behavior is not only an interaction between the tire and the pavement but is also influenced by environmental factors. Environmental factors such as temperature, rain, snow, and contaminations all affect tire–pavement contact behavior. Researchers have been most concerned with the effects of various weather and contaminations on contact behavior.

### 3.3.1. Climatic

Both tires and pavement are affected by the surrounding environment, and climate is one important environmental variable [61,66]. Pavements are affected by the climate, and there are various states such as dry, wet, ice, etc., that will make a significant difference in tire–pavement contact. Zheng et al. [67] used a combination of field measurements and finite element simulations to analyze tire–pavement adhesion properties in the dry,

wet, lubricated, and ponding states of a pavement. It was found that under the dry road state, when the slip rate was around 15%, the adhesion coefficient reached the peak value, while the adhesion coefficient peaked at a slip rate of 11.5% for wet pavement. Gerthoffert et al. [68] analyzed friction in four states—dry, wet (due to water), compacted snow, and ice. By constructing a model consisting of tire, pavement, and contaminant components, it was found that friction varied greatly between states. Wies et al. [69] analyzed pavement skid resistance, noise, and rolling resistance based on tire imprints under winter and summer environmental conditions. Tires should be designed according to seasonal differences to improve driving. Contact behavior can be affected by temperature, which is directly related to climate. Tang et al. [70] developed a tire–pavement thermal–mechanical interaction model to research the effect of temperature on pavement skid resistance. They found that higher environmental temperatures led to lower skid resistance and higher braking distances. Hernandez et al. [71] used the finite element method to analyze the effect of temperature and speed on tire contact area, deflection, and three-dimensional contact stresses. It was found that the effect of temperature on contact stress is greater than that of vehicle speed. In summary, it can be seen that the climate has a significant effect on tire–pavement contact behavior: Higher temperatures increase the tire–pavement contact area and reduce the contact stress; and a wet pavement will reduce the contact area due to the lifting effect, thus affecting vehicle braking performance.

### 3.3.2. Contaminants

The range of contaminants is broad, including rain and snow as previously mentioned. Gerthoffert et al. [68] constructed a model to study the problem of the friction coefficient degradation of contaminated pavements to establish the relationship between contaminants and friction coefficients. The friction performance of four pavement states (dry, wet, snow, and ice) was compared. The results show that their constructed model fitted the experimental data from tire and pavement friction measurement devices. Contaminants also include rubber deposits, dust particles, jet fuel, oil spills, etc. [72]. Pérez-Fortes and Giudici [73] noted that contaminants are extremely important for pavement performance and categorized contaminants into four categories based on source—those produced by vehicles, pavement embankment deterioration, road equipment, and maintenance operations. Contaminant sources are not limited to these four categories, however; for example, the surrounding soil and plants can also produce contaminants that affect pavements. Lantieri et al. [74] analyzed the effect of contaminants spilled from vehicle accidents on the skid resistance of pavements. Tire–pavement contact is bound to change when a pavement is contaminated. Thus, contamination can also affect tire–pavement contact behavior. Thorpe et al. (2008) summarized non-exhaust emission contaminants, which are the main type of contaminants affecting pavements and are known to be increasing [75]. Pomoni et al. [76] found, experimentally, that loose contaminants reduce the tire–pavement contact area. In addition, debris from worn pavement markings, snowmelt residue, and other contaminants can affect the contact behavior of tires with the road surface, affecting driving performance [77,78].

In summary, it can be seen that the tire, pavement, and environment have an important influence on the contact area and contact stress, while tire–pavement contact will also react to the tire and pavement. Therefore, analyses should be carried out with the coupling of multi-physical fields such as stress, fluid, and temperature. This can better reflect the actual tire–pavement contact behavior. However, this is difficult, so researchers have mostly analyzed tire–pavement contact behavior through experimental and simulation research.

## 4. Experimental Research

### 4.1. Measurement Methods

Experimental research was initially only able to analyze the footprint between tires and pavements using carbon paper, and so there was less experimental research conducted on tire–pavement contact behavior. With the advancement of measurement methods and

equipment, more measurement methods are now available, meaning that tire–pavement contact behavior research is gaining widespread attention [13,37]. Measurement methods mainly include the imprinting, pressure-sensitive, pressure sensor, and optical methods. The common tire–pavement contact measurement methods are summarized in Table 1.

**Table 1.** Measurement methods for tire–pavement contact behavior.

| Measurement Methods | Principle Testing Methods | Features | Equipment View |
|---|---|---|---|
| Imprinting method — Carbon paper [22] | Break of microcapsules due to pressure; colorless hidden dyes spill out and become colored by acid | Easy to operate; low cost; only applicable for measuring contact area |  |
| Imprinting method — Pressure plates [79] | Analysis of pressure magnitude and distribution based on the size of the imprint produced by the conical particles | Very low accuracy; can measure stress distribution |  |
| Pressure-sensitive method — Pressure-sensitive film [45] | Color intensity is proportional to the applied pressure | Lower accuracy; non-reusable |  |
| Pressure-sensitive method — FUJI pressure film [1] | Same as pressure-sensitive membranes | Measurement accuracy up to 0.125 × 0.125 mm; non-reusable |  |
| Pressure sensor method — Piezoresistive sensors [35] | Electrical resistance varies proportionally to the contact pressure | Easy to operate; intelligent; accuracy of 1.1 mm × 1.1 mm |  |
| Pressure sensor method — Capacitive sensors [20] | Capacitive contact principle | Good flexibility; high sensitivity; can measure dynamic contact |  |
| Pressure sensor method — Piezoelectric Sensors [80] | Piezoelectric effect | High reliability; high sensitivity; high dynamic measurement accuracy |  |
| Pressure sensor method — Stress-In-Motion system [81,82] | Multiple principles | Measures lateral and longitudinal forces; low accuracy |  |
| Optical method — Light absorption [83] | Based on the total reflection of light | High resolution; frequent calibrations required |  |

The imprint method is the earliest tire–pavement contact test method. Researchers have been conducting skid resistance and rolling resistance studies by analyzing tire footprints as early as the 20th century [84]; however, this method cannot measure pressure levels. Yu et al. [79] used a vulcanized rubber plate with conical particles and a calibrated pressure–imprint diameter relationship curve to obtain pressure data; However, the accuracy of this method is very low and does not reflect the contact between the tire and the pavement. The optical method also has the problem that the pavement is not realistically

represented [85]. This method generates images based on the phenomenon of frustrated total internal reflection in the pressure area and processes the images to show the contact area as a color contour map of pressure, which creatively introduces optical technology into the study of tire–pavement contact behavior, but this can only be used to carry out static contact studies. Liang et al. [83] improved the method by addressing the influence of ambient light by optimizing the test bench. Consequently, they were able to measure not only static pressure distributions but also pressure under dynamic conditions.

The pressure-sensitive method and flexible pressure sensor method can truly reflect tire–pavement contact behavior. Marshek et al. [45] attempted to test the pressure distribution of tire–pavement contact using pressure-sensitive films, which consist of two layers of film—one containing capsules of developer and the other providing a developing layer on which the developer acts. Capsules are of various sizes corresponding to different breakage pressures. The pressure distribution is obtained by counting the distribution of different breakage capsules. Subsequently, FUJI Japan developed a commercial film and pressure image analysis system based on this principle, and the accuracy and intelligence of this technique were greatly improved when used in pavement research [1,86]. As several manufacturers have developed sophisticated flexible pressure sensor products, researchers have begun to use these products to obtain pressure distribution data during tire–pavement contact to build more accurate tire–pavement contact models, measure tire grounding performance, pavement skid resistance, etc. [13,20]. The method is easy to operate, can be controlled by a computer program, and has high accuracy, but pressure sensors are costly and easily damaged. A future development direction is to improve the measurement accuracy of pressure sensors and make them more applicable to various test environments.

### 4.2. Current Status and Trends in Measurement Methods

In summary, it can be seen that with the improvement in the accuracy of test equipment and system improvement, several measurement methods can be applied in pavement performance and tire performance studies. However, there are different problems with each technique at this stage. The first is the cost issue; except for the imprinting method, other methods are relatively expensive. Therefore, now and in the future, there are likely to be many studies that still use the imprinting method to calculate contact stress and contact area [22]. The second issue is that of dynamic acquisition. The pressure-sensitive method has been continuously updated (from an initial accuracy of a few millimeters to $0.125 \times 0.125$ mm) and is capable of converting pressure into a pressure cloud diagram. This has great advantages with respect to measurement accuracy and the visualization of results [1]; however, the inability for dynamic continuous acquisition remains a major limitation. At the same time, this method suffers from the problem that high accuracy and range cannot be realized. In general, a key development direction is obtaining high-precision data under static conditions. The third problem is the inability to reflect the true pavement surface. The optical method mostly ignores the real pavement and is limited to tire performance studies. The pressure sensor method is available for static and dynamic tests. The United States Tekscan company, Canada Xseneor, and others have developed mature, flexible sensor equipment. This technology is simple to operate, has computer program control, provides high accuracy, and, in recent years, has shown continuous improvement, although the cost of these pressure sensors is high and they are easily damaged. The development of high-accuracy, low-cost, standardized equipment that can measure dynamic contact is a trend of the future.

### 5. Theoretical Research

#### 5.1. Mathematical Analysis of Tire–Pavement Contact Behavior

The study of tire–pavement contact behavior is not limited to experimental studies. Many researchers have constructed different contact models and carried out analyses [87–89]. One of the most classic contact models is the elastic contact model proposed by Hertz, which is also the basis for many subsequent classic models [90]. For the first time, Hertz treated

the elastomer as a quadric and analyzed its deformation and contact problems, as shown in Figure 9. The contact radius and contact stress can be calculated directly from the model, as shown in Equations (4) and (5).

$$a = \left(\frac{3WR}{4E^*}\right)^{1/3} \tag{4}$$

$$P(r) = \frac{3W}{2\pi a^2}\left[1 - \left(\frac{r}{a}\right)^2\right]^{1/2} \tag{5}$$

where $W$ is the applied load; $R$ is the compound curvature, $\frac{1}{R} = \frac{1}{R_1} + \frac{1}{R_2}$; $E^*$ is the contact modulus, $\frac{1}{E^*} = \frac{1-v^2}{E_1} + \frac{1-v^2}{E_2}$; $E$ is the elastic modulus; $v$ is the Poisson ratio; $a$ is the radius of the contact circle; $P(r)$ is the contact pressure; and $r$ is the distance from the point to the contact center.

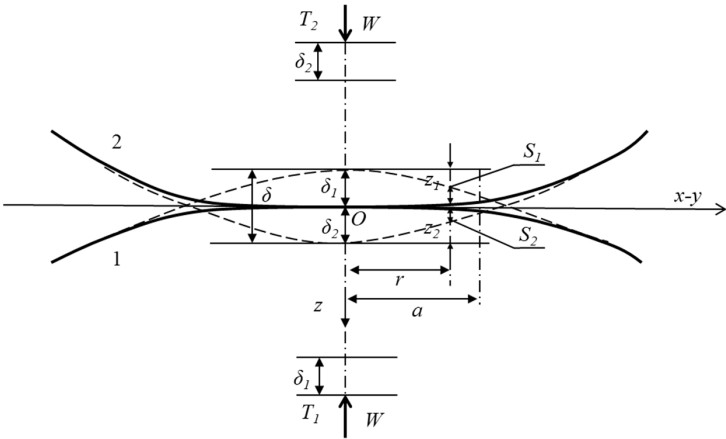

**Figure 9.** Frictionless static contact of two convex bodies.

Subsequent research based on this model adjusted the geometric parameters to obtain a model of a cylinder and an elastic half-space, as shown in Figure 10, and the contact radius and contact stress, as shown in Equations (6) and (7). Wu [91] used this model to analyze the effect of tire and pavement parameters on tire–pavement contact behavior.

$$a = \left(\frac{4FR}{E^*\pi}\right)^{1/2} \tag{6}$$

$$P(x) = \frac{2F}{\pi a^2}\left[a - x^2\right]^{1/2} \tag{7}$$

where $x$ is the distance from the point to the contact center; $L$ is the length of the cylinder; $F$ is the line load; and $R$ is the radius of the cylinder.

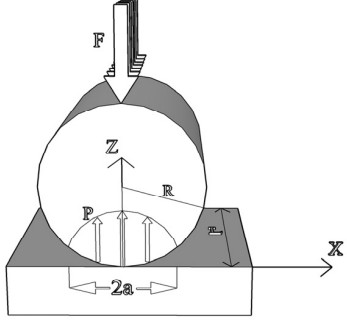

**Figure 10.** Interaction of cylinders and elastic half-space bodies.

In 1966, Greenwood and Williamson [92] proposed a mixed elastic and elasto-plastic contact model between rough and smooth surfaces based on statistical analysis, i.e., the G–W model, as shown in Figure 11. Their results showed that the actual area of objects in contact with each other, the number of micro-convex bodies, and the load generated by the contact are all related to the probability density function of the height of the surface form profile. The actual contact area is shown in Equation (8), and the actual average pressure is shown in Equation (9), where $p(z)$ is a function of the probability density.

$$A_{re} = n\pi R(z-d) = N\pi R \int_d^\infty (z-d)p(z)dz \tag{8}$$

$$p_r = \frac{4NE^*R_p^{1/2}}{3A_{re}} \int_d^\infty (z-d)^{3/2}p(z)dz \tag{9}$$

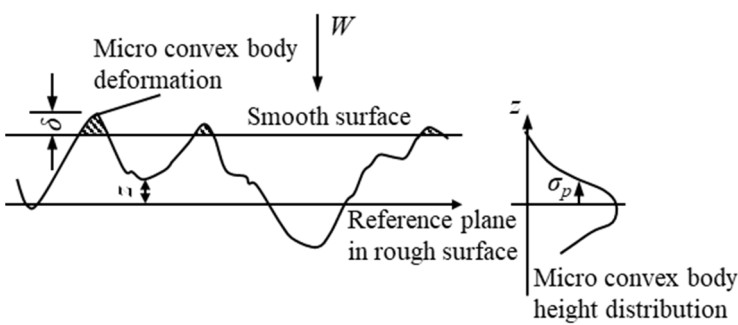

**Figure 11.** Contact between a rough surface and a smooth surface.

To further improve the accuracy of the tire–pavement model, Majumdar and Bhushan [93] introduced the fractal geometry theory and proposed the M–B fractal contact model using the W–M fractal function. This model, compared with the G–W model, includes fractal characteristics of the rough surface and can provide a quantitative representation of the relationship between surface roughness and the true contact area and the elastic and plastic contact area, respectively, as well as reflecting the relationship between load and contact area. Subsequently, Persson [94] proposed the surface fractal friction theory, which is based on the fractal characteristics of the pavement surface and the power spectrum density to model the tire–pavement contact. This model is widely used for tire–pavement friction analysis and reflects the skid resistance of the pavement by calculating the dynamic friction factor, which is calculated as shown in Equation (10). The contact state is shown in Figure 12, where $C(q)$ is the road surface spectral density; and $p(q)$ and $G(q)$ are the fractal characteristic functions, where q is the wavelength and is related to the amplification factor $\zeta$. Persson analyzed experimental data on tire–pavement contact using Persson contact mechanics and rubber friction theory with better results [95,96].

$$\mu = \frac{1}{2} \int_{q_L}^{q_1} dq\, q^3 C(q)p(q) \times \int_0^{2\pi} d\phi \cos(\phi) \cdot \mathrm{Im} \frac{E(qv\cos\phi)}{(1-v^2)\sigma_0} \tag{10}$$

In summary, it can be seen that the Hertz contact model considers surface contact as elastic contact and single-point contact, and the model is used in tire–pavement contact studies to oversimplify the characteristics of the tire and the pavement. The G–W contact model takes into account the textural properties of the contact surface but assumes the micro-convex body as a uniform semicircle, which is different from the real situation. Meanwhile, both the Hertz contact model and the G–W contact model ignore the interactions of different contact areas, while Persson's model of tire–pavement contact takes into account the scale-dependent deformation response of rubber to surface roughness to make up for this deficiency. Furthermore, the friction theory system of this model fits better with the actual situation and is widely used in the study of the pavement skid-resistance mechanism.

However, this model assumes the conservation of energy dissipated by tires and work done by friction, ignoring energy transfer and dissipation. Therefore, further refinement is still needed.

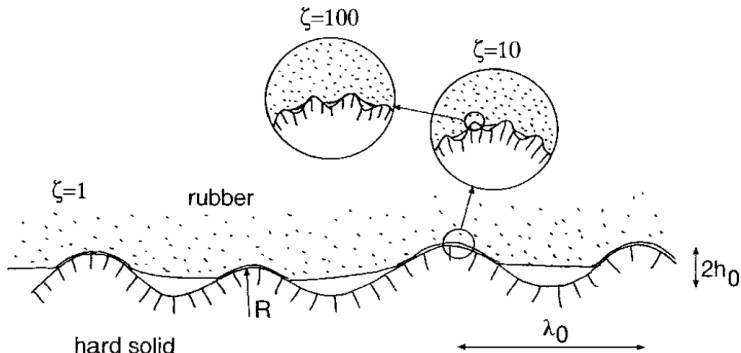

**Figure 12.** Rough pavement and tire contact.

*5.2. Numerical Simulation Analysis of Tire–Pavement Contact Behavior*

Although various tire–pavement contact models are available to analyze tire–pavement contact behavior and even the influence of pavement surface properties on the contact behavior, these models all assume the geometry and height distribution of a micro-convex body while ignoring the interaction between different contact zones; therefore, the usability of these models is very limited. With the development of computer technology, some researchers have begun to simulate tire–pavement contact behavior on computers and calculate tire–pavement contact stresses and contact areas via numerical simulation [14,60,97]. The modeling process is shown in Figure 13; contact behavior analysis is carried out by constructing models of tires and road surfaces, and some models even consider the water factor and can carry out water-wading calculations.

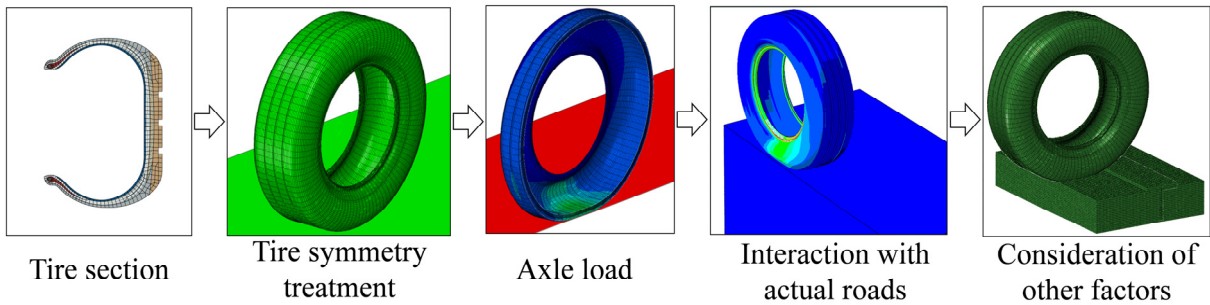

**Figure 13.** The modeling process.

Wang et al. [46] simulated tire–pavement interactions to predict contact stresses at static and various rolling conditions, using an arbitrary Lagrangian–Eulerian formulation for the steady-state rolling process of tires, and their model results were consistent with previous measurements. Their analysis results show that the non-uniformity of vertical contact stress decreases with the increase of load and increases with the increase of expansion pressure. However, vehicle-maneuvering behavior significantly affects the tire–pavement contact stress distribution. Oubahdou et al. [13] used a semi-analytical model to predict vertical contact stresses at the tire–pavement interface under different loads and inclination angles and compared the experimental measurements with Tekscan flexible sensors to confirm the reliability of the numerical analysis. Reynaud et al. [98] used the Hertz model for tire–pavement contact analysis and obtained a tire grounding pressure model using the boundary element method and semi-analytical method. Using their model, the contact differences were analyzed for different tire inflation pressures and different loads, and the contact stresses measured by using a Stress-In-Motion device were compared with the

simulation calculation results. It was shown that the model could accurately analyze the tire–pavement contact behavior. Zheng et al. [3] established a three-dimensional tire–asphalt pavement contact model using the finite element method. The effects of internal inflation pressure and wheel load on the tire–pavement contact characteristics were explored. Their results show that when a tire is under emergency braking, the shoulder position and the center region of the tread are more prone to wear, and the contact pressure in the center region is the largest. The average contact pressure when the tire was in the ABS condition increased by 8.4% and the contact area decreased by 7.7% compared to a static load. In summary, it can be seen that the use of numerical simulation has improved the analysis of tire–pavement contact behavior.

*5.3. Current Status and Trends of Theoretical Research*

At this stage, numerical simulation techniques mostly construct tire–pavement contact models by analyzing pavements and tires, with little consideration of environmental factors; only a relatively few studies have analyzed the effects of temperature and wetness on tire–pavement contact. Bai et al. [99] established a three-dimensional finite element model based on non-uniformly distributed tire–pavement contact pressure, full interfacial layer bonding conditions, and viscoelastic properties of an asphalt layer. According to this model, it was found that higher temperatures could lead to a smoother stress distribution on the wheel track, and the top of the asphalt concrete pavement showed a tendency to become damaged under the shear stress in the vertical direction as the temperature increased. Zheng et al. [67] analyzed the adhesion characteristics of the tire–pavement contact interface using numerical simulation and revealed tire–pavement adhesion characteristics under different pavement conditions. The adhesion coefficient of tire–pavement increased with the increase of MPD value in both dry and wet conditions, but the rate of change was greater under wet pavement conditions. Tires and pavements are subject to complex environmental conditions, with not only temperature changes occurring in the pavement but also dust and weather extremes affecting the tire–pavement contact behavior. Therefore, the analysis of tire–pavement contact behavior under multi-physical field coupling should be carried out; only by constructing contact models for tires, pavements, and the environment can the contact process between tires and pavements be truly described. However, this puts higher requirements on model construction and computational power. Therefore, currently, existing simulation analyses still have many shortcomings.

## 6. Analysis Based on Tire–Pavement Contact Behavior

The purpose of tire–pavement contact behavior research is to analyze pavement performance and tire performance. The pavement engineering and tire fields have research needs, but older testing methods are difficult to employ and offer low accuracy. Researchers can only simulate and mathematically calculate tire–pavement contact behavior to analyze pavement and tire performance. With the development of equipment, researchers have gradually begun conducting contact-behavior tests to evaluate road performance and tire performance. Currently, there are many types of pavement-performance and tire-performance analyses based on tire–pavement contact behavior, such as skid resistance, noise, pavement damage, rolling resistance, etc.

*6.1. Skid Resistance*

Firstly, pavement skid resistance has been previously studied. Peng et al. [100] developed a skid-resistance simulation framework based on a finite element method using in situ 3D pavement texture and skid resistance data. The relationship between each contact feature and the interface friction parameters was quantified by principal component analysis and regression analysis. The results of the study provided better friction parameter inputs for finite element simulations. Najafi et al. [101] concluded that friction is reduced and vehicle crash rates are increased under wet weather conditions due to the water on the pavement reducing the contact area between the tire and the pavement. Yun et al. [5] also

realized the problem of friction change and studied the effect of contact area on the friction coefficient. The relationship between pavement texture and contact area was investigated; the surface textures of 29 pavements were scanned; and the skid resistance of 23 of them and the tire–pavement contact area of some were also measured. The results showed that the contact area is influenced by the roughness of the pavement; the contact area decreases as a power function of the root-mean-square surface height; and the low-speed tire–pavement friction coefficient under dry conditions is not influenced by the real contact area, while the friction coefficient under wet conditions is weakly positively correlated with the real contact area. Yu et al. [102] used a pressure-distribution-mapping system, FPD-8010E, and a tire–pavement dynamic friction analyzer to test the contact area and friction coefficient of different pavements. They found a close linear correlation between the actual tire–pavement contact area and the dynamic friction coefficient (DFC), as shown in Figure 14. It can be seen that the DFC gradually increases as the actual area of contact increases, but this correlation varies with pavement type and aggregate type. Skid resistance is certainly related to tire–pavement contact behavior, but it is also influenced by many other factors, such as aggregate type [103], pavement gradation [104], etc. In summary, the analysis of pavement skid resistance requires a comprehensive understanding of tire–pavement contact behavior.

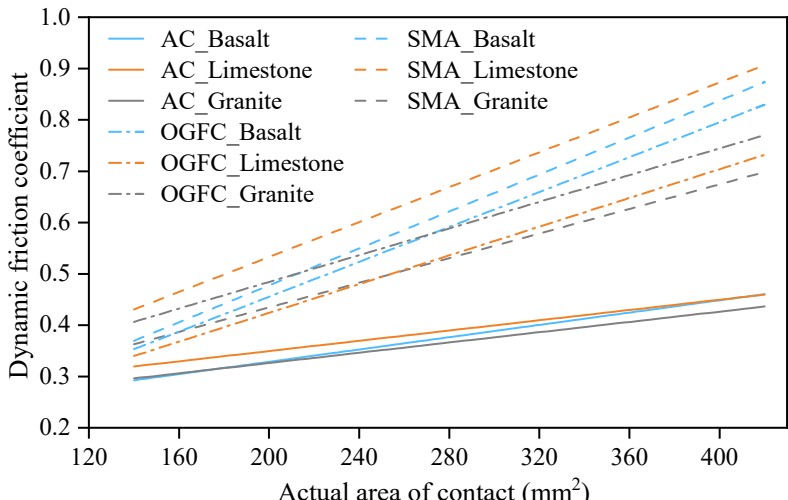

**Figure 14.** Fitting graph of the dynamic friction coefficient to the actual contact area.

### 6.2. Noise

In addition to pavement skid resistance studies based on contact behavior, researchers have also conducted noise studies. Ding [27] established a coupled finite element and boundary element analysis method (FEM/BEM) for tire–pavement interaction noise simulation to quantify the effect of pavement characteristics on the generation and propagation of tire vibration noise. It was found that tire–pavement noise increased with increasing texture level and that overall noise decreased with increasing porosity at the same texture level. Noise research has mostly used a combination of noise and stress equipment to analyze the effect of contact behavior on noise, as shown in Figure 15. Cesbron et al. [105] used a dynamic pressure-measurement system and a close-proximity (CPX) noise-measurement system to measure the contact force and noise levels for a slick tire rolling on six different pavement surfaces between 30 and 50 km/h. The results showed that contact forces and close-proximity noise measured at 30 km/h were correlated. Later, Cesbron et al. [106] summarized a relationship function between noise level and contact force level based on tire–pavement contact data and CPX rolling noise measurements, as shown in Equation (11). However, this relationship is valid only at low frequencies (315–1250 Hz for a slick tire and 315–800 Hz for a patterned tire). Huang et al. [107] used pressure film to measure the open void ratio at the tire–pavement contact interface and judged pavement noise according to

the open void ratio. Their study showed that as the open void ratio increased, the noise between the tire and the pavement decreased.

$$L_N^e = a_i L_F^e(f_i) + b_i \tag{11}$$

where $L_N^e$ is the contact force level; $L_F^e(f_i)$ is the noise level; and $a_i$ and $b_i$ are the coefficients of the linear regression using a statistical method.

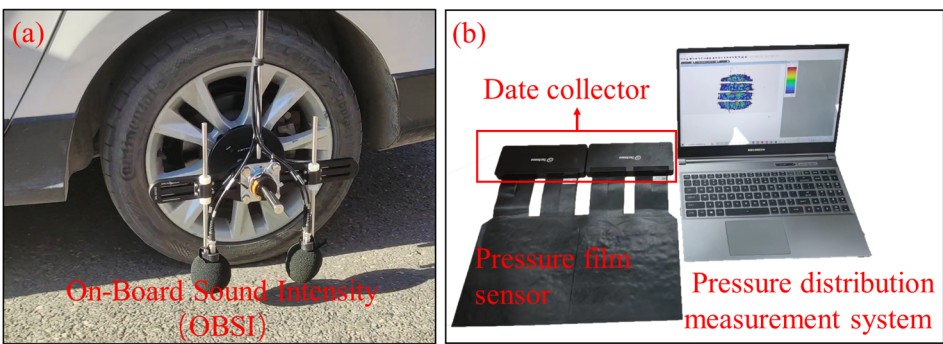

**Figure 15.** Noise test and contact stress test equipment: (**a**) On-Board Sound Intensity (OBSI) test method; (**b**) Pressure distribution measurement equipment.

### 6.3. Pavement Damage

Pavement damage is also closely related to tire–pavement contact behavior, and uneven tire–pavement contact stresses play an important role in the sprouting of top-down cracks. Studies of this phenomenon point out that high surface stresses may be the origin of the top-down crack creation. Manyo et al. [108] constructed a simulation model of tire–pavement contact pressure and used flexible sensor data to verify its reliability. They then analyzed the effect of tire–pavement contact behavior on the pavement structure. Their analysis showed that the maximum tensile strain was capable of causing top-down cracking. This strain is higher than the level considered in current pavement design models [109]. Hajj et al. [42] used a quartz sensor array to measure tire–pavement contact stress distributions and calculated pavement responses to predict bottom-up fatigue cracking using the Mechanistic–Empirical Pavement Design Guide (MEPDG). It was found that equivalent circular and square contact stress distributions overestimated bottom-up fatigue cracking by 4% to 23%, and the elliptical stress distributions resulted in similar or underestimated fatigue cracking by up to 17%. Contact behavior does not only affect pavement cracking but is also related to rutting damage. Ronald et al. [110] tested contact stresses with the Vehicle–Road Surface Pressure Transducer Array System and found that high contact stresses tend to cause rutting. Moazami and Muniandy [111] found that the actual contact area and stresses were not different from the design, which would cause the pavement to be highly susceptible to rutting damage. In summary, it can be seen that pavement damage is closely related to tire–pavement contact behavior. Understanding the true contact stresses will help design economical and durable pavements.

### 6.4. Rolling Resistance

Tire–pavement contact tests also have an important role in the field of automotive tires, where contact characteristics are closely related to rolling resistance. Guo et al. [15] established a three-dimensional tire–pavement model to analyze the changing characteristics of the contact stress and the relationship between the rolling resistance and tire–pavement operating conditions. Their results show that the strength of the relationship between transverse and longitudinal contact stresses is related to rolling conditions, and a method was proposed that can be used to predict the relationship between truck–bus tire conditions and rolling resistance. Kawakami et al. [7] reported tire–pavement pressure distributions using a pressure-measuring film and analyzed the relationship between rolling resistance

and pavement contact characteristics. Their results for the correlation between the area percentage and the rolling resistance coefficient at 20 km/h are shown in Figure 16. It can be seen that the distribution of the contact pressure correlates with the rolling resistance coefficient at low vehicle speeds, and so the contact pressure distribution can be used to indirectly evaluate the rolling resistance. Liang Chen et al. [112] used the Tekscan pressure-measurement system to obtain the grounding pressure distribution, which was used to construct a parametric evaluation system expressing geometric and mechanical information in the ground area. Their results showed that the regression equation fitted better when tire–pavement characteristic parameters were used to evaluate the tire rolling resistance, and the maximum relative fitting error was only 2.367%; the corresponding equation is given as Equation (12).

$$f_1 = 1.324 - 0.028\overline{\Delta Z'_{\mathrm{I}}} + 0.021E_1 - 0.031x_9 - 0.038\overline{\Delta Z'_{\mathrm{II}}} + 0.100\sigma E_{y_{\mathrm{II}}} + 0.020E_{\mathrm{IV}} - 0.294x_{14} \tag{12}$$

where $f_1$ is the rolling resistance coefficient and the other parameters represent the outer tire shoulder area and transition area tire–pavement contact characteristics.

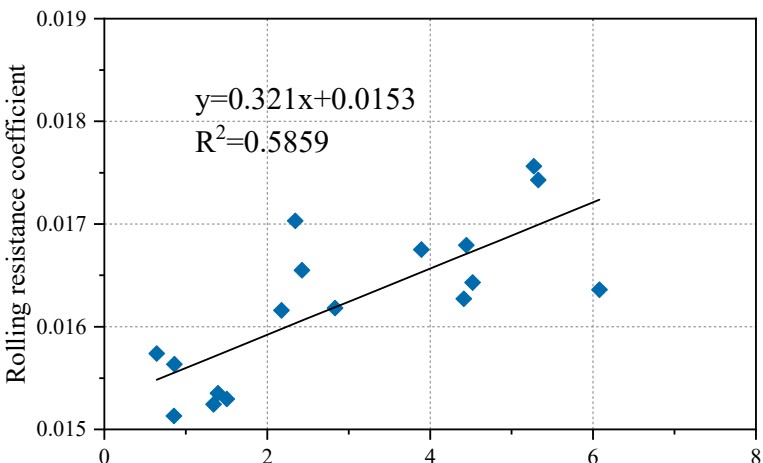

**Figure 16.** Relationship between the rolling resistance coefficient and the percentage of the area [7].

In addition to the above-described tire and pavement performance studies, contact behavior studies can also be used to analyze structural stresses in pavement [108], pavement durability [113], and other issues [42]. Therefore, it is crucial to define real tire–pavement contact behaviors. When contact behaviors are identified as such, road design and tire design can be optimized based on contact stresses, the extent of pavement damage by stress can be determined, changes in pavement performance can be analyzed based on changes in contact characteristics, and pavement damage processes can be understood. In summary, many pavement- and tire-performance analyses can be performed based on contact behavior.

## 7. Concluding Remarks

This paper has reviewed the research progress on tire–pavement contact behavior in recent years from the perspectives of tire–pavement contact, contact behavior experimental research, theoretical research, and analyses based on tire–pavement contact behavior. It was found that tire–pavement contact behavior research suffers from low experimental accuracy, high costs, and the limited applicability of contact models. Therefore, future in-depth research should be conducted mainly on the following aspects:

(1) Defining tire–pavement contact characteristics. Tire–pavement contact characteristics are mainly geometric and mechanical. There is a lack of definitions for specific contact

characteristics in the pavement field, and the influences of different characteristics on pavement performance are poorly understood and require further research.

(2)    Many factors influence tire–pavement contact characteristics. Tire–pavement contact behavior in complex environments is inherently more complex, especially when rain, dust, and pollutants are present. This means that the comprehensive analysis of geometric characteristics, mechanical characteristics, and environmental factors remains limited and will be the focus of future research.

(3)    The existing geometric characteristic test lacks the contact depth test and is limited to imprint measurement, and the mechanical feature test is only a vertical stress test, excluding transverse and longitudinal stresses. Tire–pavement contact tests are mostly static and non-standardized, and it is a future trend to develop high-precision, low-cost, and standardized instruments that can comprehensively measure dynamic contact stresses.

(4)    In terms of theoretical research, finite element numerical simulation has more development potential than traditional tire–pavement contact calculation model research. Tire–pavement models need to be more realistic and coupled with multiple environmental factors. Model calculations will always differ from reality, and the more realistic the tire path model, the more complex the calculations.

(5)    The performance of most pavement changes over time, and this change process is directly related to tire–pavement contact behavior while also being influenced by the pavement environment. Therefore, pavement performance evaluation indexes need to be established in the future based on tire–pavement contact characteristics.

**Author Contributions:** Conceptualization, Y.M. and Z.G.; investigation, Y.M., C.L. and Z.G.; writing, Z.G. and C.L. All authors have read and agreed to the published version of the manuscript.

**Funding:** This work was jointly supported by the National Natural Science Foundation of China (No. 51978048) and the China Scholarship Council (No. 202306460065).

**Data Availability Statement:** Data are contained within the article.

**Conflicts of Interest:** The authors declare no conflicts of interest.

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
