# Peer review of "Review of Research on Tire–Pavement Contact Behavior"

_coatings, doi:10.3390/coatings14020157_

Round 1
Reviewer 1 Report
Comments and Suggestions for Authors
This is a very interesting manuscript that addresses important issues related to the latest progress in tire-pavement contact behavior.
The paper is well structured and clearly written.
Descriptions of the references should be extended. For example: “Tire-pavement contact involves tribology, acoustics, contact mechanics, tire dynamics and other multidisciplines, and has attracted widespread interest in the field of pavement engineering [10-12].” there are 3 references, but the reader does not know what they refer to; the same for ([1-3], [13-15], [18-20], [21-23], [26,30,31], [38-40]).
From these reasons, I think that this manuscript, after a minor revision, could be accepted to be published by the Coatings journal.
Author Response
Many thanks for your comments. We have expanded the description of references or adjusted the location of relevant references.
Reviewer 2 Report
Comments and Suggestions for Authors
Line 110. Please explain what the equations shown in the graphs in Figure 3 mean?
Line 116, 117. Please provide the definition of the abbreviations FUJI and LLW.
Line 258. In the caption to Figure 8, please add the designations “a”–“d” with a description of the images that are associated with them;
Line 619. Figure 16. “Ratio” is usually a dimensionless parameter. However, the authors use "%" as a unit of measurement for "Ratio of the area..."
Author Response
1. Please explain what the equations shown in the graphs in Figure 3 mean?
Answer: Many thanks for your comments. This equation is a relationship between tire pressure and wheel load, and is also a boundary equation with significant differences in the shape of the tire-pavement contact above and below the equation.
2. Line 116, 117. Please provide the definition of the abbreviations FUJI and LLW.
Answer: FUJI is a company name. LLW is one of the company's products.
3. Line 258. In the caption to Figure 8, please add the designations “a”–“d” with a description of the images that are associated with them;
Answer: We have adjusted the statement of the equations in the paper according to your suggestions. The revised text was marked in bule.
Revised text: Figure. 8 Asphalt and cement pavements: (a) Asphalt pavement; (b) Cement pavement; (c) Asphalt pavement texture; (d) Cement pavement texture.
4. Line 619. Figure 16. “Ratio” is usually a dimensionless parameter. However, the authors use "%" as a unit of measurement for "Ratio of the area..."
Answer: I think you're right, we replaced 'Ratio' with 'Percentage'.
Revised text: The results of the correlation between area percentage and rolling resistance coefficient at 20 km/h are shown in Figure 16.
Reviewer 3 Report
Comments and Suggestions for Authors
The paper is a review of research on tire-pavement contact behavior. It describes very important and also interesting problem. The minor remarks are as follows:
ABSTRACT:
The abstract is written properly and captures main aspects of the paper. The keywords are also selected properly.
INTRODUCTION:
The introduction presents the overall background of the problem considered. I only suggest to remove subsections 1.1 and 1.2. The uniform text looks better.
SECTION 2 - Tire-Pavement Contact Characteristics
The section is written clearly and includes both geometrical and mechanical characteristics of the tire-pavement contact.
SECTION 3:
There are a lot of subsections here and I suggest to remove subsections, like 3.2.2.
SECTIONS 4-6:
These sections are written thoroughly and carefully. There are also enriched by tables and figures which is a strong point of the paper.
CONCLUDING REMARKS:
This section captures the main conclusions associated with the work. There are written properly and in a readable way.
REFERENCES:
References are selected properly and within the scope of the paper. Their amount is also sufficient for the review paper.
Author Response
ABSTRACT:
Thank you for your recognition.
INTRODUCTION:
We have adjusted the structure in the paper according to your suggestions. I remove subsections 1.1 and 1.2.
SECTION 2 - Tire-Pavement Contact Characteristics:
Thank you for your recognition.
SECTION 3:
Answer: The reason for the more detailed division of this section is that there are many influencing factors. It is difficult to generalize, so the subsections have not been adjusted.
SECTIONS 4-6,CONCLUDING REMARKS and REFERENCES:
Answer: Thank you for your recognition.
Reviewer 4 Report
Comments and Suggestions for Authors
The paper is well-structured. The following comments shall be considered for additional improvements.
1_ Please explain in the introduction how the consideration of tire pavement interaction could help scientists and practitioners on the pavement design and analysis procedures, i.e., accuracy, etc.
2_ In section 6.1, the impact of several and alternative materials on the provided skid resistance, should be highlighted, e.g., https://doi.org/10.3390/recycling7040047
3_ Please provide current modelling limitations and future research challenges at the end of the conclusions.
4_ Please comment somewhere in the text on potential factors that could make the related stakeholders reluctant from routinely applying tire-pavement interaction, e.g., need for time and computational resources, etc.
Comments on the Quality of English Languagemoderate changes are needed
Author Response
1_ Please explain in the introduction how the consideration of tire pavement interaction could help scientists and practitioners on the pavement design and analysis procedures, i.e., accuracy, etc.
Answer:
We have added relevant explanations in the introduction based on your suggestion.
Revised text:
Therefore, the study of tire-pavement contact behavior is of great significance to im-prove pavement performance, enhance pavement life, and vehicle energy saving and environmental protection [19,20]. Meanwhile, only with a clear understanding of the interaction between the tire and the pavement, can we get real contact stress data, and pavement load distribution. Only then can we design a more economical and durable pavement, as well as carry out friction analysis.
2_ In section 6.1, the impact of several and alternative materials on the provided skid resistance, should be highlighted, e.g., https://doi.org/10.3390/recycling7040047
Answer:
We have added relevant content based on your suggestions
Revised text:
It can be seen that the DFC gradually increases as the actual area of contact increases, but this correlation varies with pavement type and aggregate type. Certainly, skid resistance is related to tire-pavement contact behavior, but it is also influenced by many factors. Such as aggregate type [103], pavement gradation, etc [104]. But in summary, the analysis of pavement skid resistance requires a good understanding of the tire-pavement contact behavior.
3_ Please provide current modelling limitations and future research challenges at the end of the conclusions.
Answer:
We have added the appropriate conclusions based on your suggestions.
Revised text:
(4) In terms of theoretical research, finite element numerical simulation has more development potential than the traditional tire-pavement contact calculation model research. Tire-pavement models need to be more realistic and coupled with multiple environmental factors. Model calculations will always differ from the real, and the more realistic the tire path model, the more complex the calculations.
4_ Please comment somewhere in the text on potential factors that could make the related stakeholders reluctant from routinely applying tire-pavement interaction, e.g., need for time and computational resources, etc.
Answer: There are a number of reasons for the reluctance to carry out tire-pavement interaction analyses, for the reasons for the unwillingness to conduct experimental research see 4.2. At the same time we have added some content in 5.3 to answer the reasons for reluctance to conduct simulation research.
Revised text:
Therefore, the analysis of tire-pavement contact behavior under multi-physical field coupling should be carried out, and only by constructing the contact model of tire, pavement and environment can the contact process between tire and pavement be truly described. However, this puts higher requirements on model construction, computational power, and therefore existing simulation analysis still has many shortcomings.
Reviewer 5 Report
Comments and Suggestions for Authors
Accept in present form.
Author Response
Thank you for your recognition.
Round 2
Reviewer 2 Report
Comments and Suggestions for Authors
Line 117-118. Please provide the definition of the abbreviations FUJI and LLW in the manuscript, if it possible.
Author Response
We have added definitions of abbreviations according to your suggestions. The revised text was marked in orange.
Revised text:
Yu et al. (2021) conducted a static tire-pavement contact test using Fujifilm Holdings Corporation (FUJI)pressure film paper with super low pressure (LLW)type.
Reviewer 4 Report
Comments and Suggestions for Authors
None.
Comments on the Quality of English LanguageMinor checks.
Author Response
We have checked the English language and made changes based on your suggestions. The revised text was marked in violet.